

# Assessment of the recovery and photosynthetic efficiency of *Breviolum psygmophilum* and *Effrenium voratum* (Symbiodiniaceae) following cryopreservation

Joseph K. Kihika[1,2], Susanna A. Wood[2], Lesley Rhodes[2], Kirsty F. Smith[2,3], Juliette Butler[2] and Ken G. Ryan[1]

[1] Department of Biological Sciences, Victoria University of Wellington, Wellington, New Zealand
[2] Cawthron Institute, Nelson, New Zealand
[3] Department of Biological Sciences, University of Auckland, Auckland, New Zealand

## ABSTRACT

Many strains of Symbiodiniaceae have been isolated and their genetics, taxonomy, and metabolite production studied. Maintaining these cultures requires careful and regular sub-culturing that is costly with a high risk of species contamination or loss. Cryopreservation is a viable alternative for their long-term storage; however, there is uncertainty as to whether cryopreservation impacts the photosynthetic performance of Symbiodiniaceae. We investigated the growth rates and photosynthetic efficiency of two species, *Breviolum psygmophilum* and *Effrenium voratum* before and after cryopreservation. Rapid light curves (RLCs) produced using Pulse Amplitude Modulated (PAM) fluorometry were used to generate detailed information on the characteristics of photosystem II (PSII). The maximum electron transport rate (ETRmax) and the quantum yield ($F_v/F_m$) of the control (non-cryopreserved) and cryopreserved culture isolates were assessed across the growth cycle. The non-cryopreserved isolate of *B. psygmophilum* had a higher quantum yield than the cryopreserved isolate from day 12 to day 24, whereas there were no differences from day 28 to the late stationary phase. There were no significant differences in ETRmax. No significant differences were observed in quantum yield or ETRmax between the control and cryopreserved *E. voratum* isolates. The ability of cryopreserved strains to recover and regain their photosynthetic efficiency after freezing demonstrates the utility of this method for the long-term storage of these and other Symbiodiniaceae species.

## INTRODUCTION

The family Symbiodiniaceae consists of eleven formally described genera (*Pochon & LaJeunesse, 2021*) and many species (*Qin et al., 2019*) whose ecological strategies range from endosymbiotic to free-living (*Pochon & LaJeunesse, 2021*). Endosymbiotic species

Corresponding author
Joseph K. Kihika,
joseph.kihika@vuw.ac.nz

are associated with numerous marine invertebrate taxa (*Coffroth & Santos, 2005*). For example, corals get 90% of their nutrition through photosynthesis from their symbiotic Symbiodiniaceae (*Kemp et al., 2015*). These dinoflagellates also protect the corals against damaging ultraviolet radiation (*Stanley, 2006*) and some play a role in the biosynthesis of secondary metabolites (*Santiago-Vázquez, Newberger & Kerr, 2007*).

Due to on-going impacts from global warming, coral bleaching and habitat destruction, there is a need to establish a long-term preservation method for these dinoflagellates, in particular the symbiotic forms, to facilitate coral-reef restoration efforts (*Cirino et al., 2019*; *Hagedorn & Carter, 2015*; *Kihika et al., 2022a*). Maintaining many living dinoflagellate culture isolates is costly, requires regular sub-culturing of living cells into new media, and there is a constant risk of contamination from unwanted microorganisms (*Kihika et al., 2022a*; *Rhodes et al., 2006*). Cryopreservation is a technique of preserving viable biological samples such as cells, tissues or microorganisms in a frozen state over extended periods of time at relatively low cost (*Nakanishi, Deuchi & Kuwano, 2012*; *Tsai & Lin, 2012*). Multiple Symbiodiniaceae isolates have been successfully cryopreserved while some isolates from genera/clade A are unable to cryopreserve (*Kihika et al., 2022b*).

In microalgal cells, growth conditions and factors that induce stress can be monitored by assessing variability in chlorophyll *a* fluorescence (*Schreiber et al., 2002*). In photosynthetic cells, chlorophyll fluorescence analysis provides an extremely sensitive tool for examining energy metabolism and interactions between carbon and nutrient assimilation (*Oxborough et al., 2000*). Light energy absorbed by chlorophyll pigments can either be re-emitted as heat, fluorescence or absorbed for use in photochemical processes (*White, Anandraj & Bux, 2011*). The proportion of energy used for photochemical processes is inversely related to the amount of fluorescence emission from chlorophyll *a* (*Schreiber et al., 2002*). Pulse Amplitude Modulated (PAM) fluorometry allows investigations into photosynthetic processes based on the variability of chlorophyll *a* fluorescence (*Baker, 2008*; *Long et al., 2018*; *Oxborough et al., 2000*; *Petrou et al., 2008*). In phytoplankton physiology, PAM is used to provide insights into cell "photosynthetic health". It is a rapid, non-invasive, and non-destructive method that can detect sublethal effects on photosystems (*Long et al., 2018*). PAM fluorescence measurements have been widely used to assess the photochemical status of different microalgae (*Gilbert et al., 2000*; *Kromkamp & Peene, 1999*; *Suggett et al., 2003*). PAM fluorometry has not been used previously to characterize and compare the photosystem II function of non-cryopreserved and cryopreserved Symbiodiniaceae isolates.

The aim of this study was to investigate the use of PAM fluorometry for monitoring the photosynthetic efficiency after cryopreservation of two Symbiodiniaceae culture isolates from different genera to assess their recovery and performance post thawing. The two study species were *Breviolum psygmophilum,* which is symbiotic and is harbored by subtropical and temperate stony corals (*e.g.*, *Astrangia*, *Cladocora*, and *Oculina* species) from the Atlantic Ocean and Mediterranean Sea (*Lajeunesse, Parkinson & Reimer, 2012*), and *Effrenium voratum,* which is exclusively free-living (non-symbiotic) and is distributed in the Pacific and Atlantic Oceans, occurring at sub-tropical and temperate latitudes (*LaJeunesse et al., 2018*). It was hypothesized that: (a) after cryopreservation, the Symbiodiniaceae cells will have a lower photosynthetic activity as they recover from the cold temperatures due

to energy required to repair cell organelles, but the amount of fluorescence will increase as the cells recover, and (b) the changes in photosynthetic activity will differ between the free-living and symbiont species.

To address these hypotheses, cultured *B. psygmophilum* and *E. voratum* were cryopreserved with 15% dimethyl sulfoxide (DMSO) using methods previously described (*Kihika et al., 2022b*; *Kihika et al., 2022a*). After thawing, their growth was tracked, and their photosynthetic efficiency assessed using PAM fluorometry. They were compared with non-cryopreserved culture isolates during their late exponential and stationary growth phases. Establishing the health of the cells and stress levels in the Symbiodiniaceae after cryopreservation will help to determine how species recover post cryopreservation and assist in optimizing thawing processes.

## MATERIALS AND METHODS

### Cultures studied and their growing conditions

Two dinoflagellate culture isolates from the family Symbiodiniaceae: *B. psygmophilum* and *E. voratum* were obtained from the Marine Symbiosis and Coral Reef Biology laboratory at Victoria University of Wellington (New Zealand). The culture isolates had different cell sizes with the mean girdle diameter of *B. psygmophilum* $6.69 \pm 0.83$ µm and *E. voratum* $9.29 \pm 0.77$ µm. To start the cultures, approximately 100 cells mL$^{-1}$ of each isolate was used. The cultures were grown in f/2 growth medium enriched with nutrients (*Guillard, 1975*). They were maintained in sterile plastic flatbottomed vessels (70 mL Labserv, ThermoFisher Scientific, NZ) with a 12:12 light: dark cycle under 100 µmol m$^{-2}$ s$^{-1}$ photosynthetically active radiation (PAR) at 25 °C. All cultures were harvested during their late exponential phase prior to the freezing experiments.

### Rapid freezing technique

The two Symbiodiniaceae culture isolates were grown to late exponential phase and cryopreserved using the rapid freezing technique described in (*Kihika et al., 2022a*). In summary, aliquots (1 mL) of each culture isolate were pretreated with 15% DMSO until a final concentration of 1:1 dilution was obtained. The cells were equilibrated in the dark for 30 min. The treated cells were transferred into 0.5 mL cryopreservation straws (medium straws clear; Imv Technologies, L'Aigle, France) which were then sealed by plugging with coloured polyvinyl chloride powder. The straws were arranged horizontally on a metal rack measuring ($41 \times 14 \times 4$ cm, l × w × h) fitted onto polystyrene floats. The rack was then gently placed over a liquid nitrogen bath ($45 \times 30 \times 6$ cm, l × w × h) and floated for 10 min inducing rapid freezing (Fig. S1). Lastly, the straws were plunged into liquid nitrogen and transferred to a storage dewar containing liquid nitrogen and stored for one week (Fig. S1).

### Thawing procedure

The cryopreserved culture isolates were thawed using a procedure described previously (*Kihika et al., 2022a*). Briefly, the cryopreserved straws were plunged into a water bath (20 °C) for few seconds until all the ice melted. Under sterile conditions, the contents

of each straw were transferred into individual empty sterile plastic flatbottomed vessels and diluted stepwise using f/2 growth medium up to a total volume of 5 mL. After an equilibration period of 30 min in the dark, a final 5 mL of f/2 medium was added. The Symbiodiniaceae cells were kept for 24 h in the dark to recover and then for 48 h under red light. Finally, 50 mL of f/2 media was added to the culture isolates, and these were transferred into their normal growing conditions.

## Growth curve and viable cell counts

Five straws containing either *B. psygmophilum* or *E. voratum* were thawed into different plastic flatbottomed vessels with equal f/2 media volume and their growth monitored. Before subsampling for the isolates cell counts, cultures were gently resuspended using a pipette. A 100–1,000 μL aliquot was taken from each plastic flatbottomed vessel, preserved with Lugol's Iodine and a serial ten-fold dilution made using the species growth media for easier determination of the cell concentration (*Kihika et al., 2022a*). An aliquot (100 μL) of the diluted culture suspension was dropped on a boundary slide glass (S6113; Matsunami Glass, Osaka, Japan) and counted in triplicate using the ''drop on slide'' method (*Anderson & Throndsen, 2003*; *Nishimura et al., 2021*) with an inverted microscope (Olympus CK X41 Tokyo, Japan). All the fixed cells on the slide were counted at appropriate magnification and then multiplied with the dilution factor to get the cell density $mL^{-1}$. For the determination of each Symbiodiniaceae isolate growth curve, cell counts were undertaken every two days. Cell counts began three days after thawing in the cryopreserved samples, which was when normal cell division was first observed. Growth rate (divisions $day^{-1}$) was calculated using the cell density from the exponential portion of the growth curve by least squares regression (*Hashimoto et al., 2021*). Maximum growth rates were calculated for both the cryopreserved and the control culture isolates, and the highest cell density obtained during the growth period was expressed as the maximum cell density $mL^{-1}$.

## PAM fluorometry

The photosynthetic performances of both the cryopreserved and non-cryopreserved *B. psygmophilum* and *E. voratum* culture isolates were measured using a PAM fluorometer (PHYTO-EDF; Heinz Walz GmbH, Germany) equipped with a fibreoptics probe. The gain of the culture isolates was determined before measurements began and set using a corresponding healthy culture for each of the culture isolates. In each experiment, sub-samples (1.4 mL) of each culture isolate were transferred into sterile two mL vials (Agilent Technologies, Beijing, China) wrapped with an aluminium foil (Emperor, New Zealand) and then incubated in the dark for 5 min to allow the full oxidation of photosystem II (PSII) reaction centers and electron transport chain. Five replicates ($n = 5$) of each culture isolate for both the cryopreserved and non-cryopreserved were used.

Rapid light curves (RLCs) were used to generate and extract detailed information on the saturation characteristics of PSII (*Ralph & Gademann, 2005*; *White, Anandraj & Trois, 2013*). Fluorescence variables and PSII quantum yield ($F_v/F_m = (F_m - F_0)/F_m$) were measured where: $F_0$ = initial fluorescence intensity, and $F_m$ = maximal intensity under saturating light conditions, and $F_v = F_m - F_0$ (*Long et al., 2018*). The quantum yield of PSII

was then calculated as $F_v/F_m$ (*Strasser, Srivastava & Tsimilli-Michael, 2000*). Relative ETR = $F_v/F_m \times$ PAR $\times 0.5 \times 0.84$ where $F_v/F_m$ estimates the quantum yield of PSII, PAR = quantum flux density of incident photosynthetically active radiation (the actinic irradiance in μmol quanta m$^{-2}$ s$^{-1}$), 0.5 = a multiplication factor based on the assumption that 50% of the absorbed quanta are distributed to PSII, 0.84 = An assumption that an average of 84% of incident PAR is absorbed by the chlorophyll in the cell (*Beer et al., 1998*; *Schreiber, 2004*). All PAM data and generation of RLCs were done using Phyto-Win software (v 1.45). The rapid light curves were generated by calculating maximum electron transport rate (ETRmax) through PSII for each level of actinic light as a function of irradiance (PAR). The measurements from the two different culture isolates were taken on days 12, 16, 20, 24, 28, 32 and 36 during their normal incubation period from late exponential to stationary growth phases. The photosystem II (PSII) maximum quantum yield and the maximum electron transport rate (ETRmax) of the cryopreserved and the control isolates were assessed and compared during the two main time points in their growth curve. First, during late exponential phase (day 12) when the Symbiodiniaceae isolates had the highest photosynthetic activity and second, during stationary phase (day 36) when they had low photosynthesis levels.

## Statistical analysis

All statistical analyses were performed in R (version 4.0.3; *R Core Team, 2020*). Nonparametric Wilcoxon tests were applied to compare maximum growth rates, maximum cell density, quantum yield and ETRmax between the control and cryopreserved culture isolates of *B. psygmophilum* and *E. voratum*. Linear regression was used to explore relationship between quantum yield and the ETRmax over the experiments. All data were expressed as mean ± standard deviation and *p*-values less than 0.05 were considered statistically significant.

# RESULTS

## Cryopreservation of Symbiodiniaceae culture isolates

Both the *B. psygmophilum* and *E. voratum* culture isolates were successfully cryopreserved using the rapid freezing method. *Breviolum psygmophilum* had a higher survival rate post-thawing (cell viability 82.9 ± 4.3%) compered to *E. voratum* (58.5 ± 5.2%).

## Growth curve of the cryopreserved and control culture isolates
### *Breviolum psygmophilum*

The maximum growth rate for non-cryopreserved *B. psygmophilum* cultures was 1.14 divisions day$^{-1}$ and the maximum cell density 6. 3$\times 10^4$ cells mL$^{-1}$ after 16 days in culture. The maximum growth rate for the cryopreserved cells was 0.86 divisions day$^{-1}$ and the maximum cell density was 9. 3$\times 10^4$ cells mL$^{-1}$after 34 days (Fig. 1). There was no significant difference in the maximum growth rate and maximum cell density between the cryopreserved and the non-cryopreserved culture isolates (Wilcoxon test; maximum growth rate *p*-value = 0.0947, maximum cell density *p*-value = 0.0947). There were no significant differences in the growth rates of the cryopreserved and the non-cryopreserved isolates during the late exponential growth phase (*p*-value = 0.060).

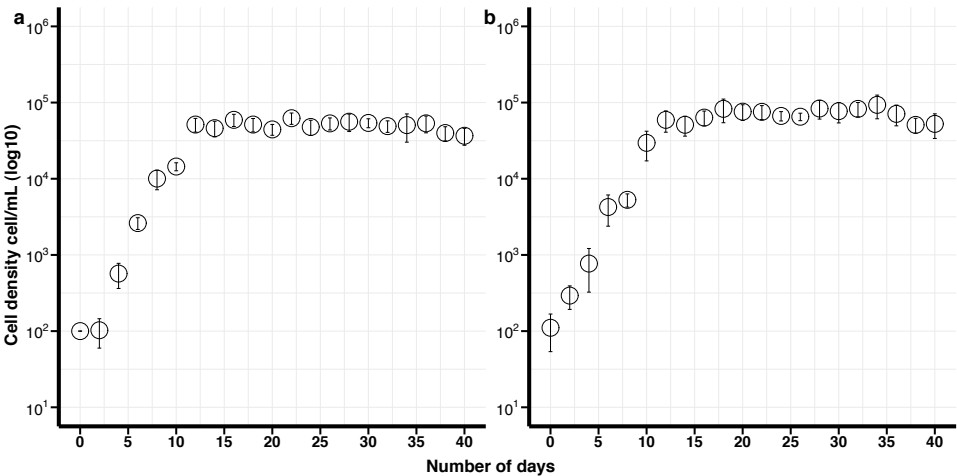

**Figure 1 Growth curves of *Breviolum psygmophilum* culture isolates with standard deviation bars (SD).** (A) Control (non-cryopreserved) culture, (B) cryopreserved culture.

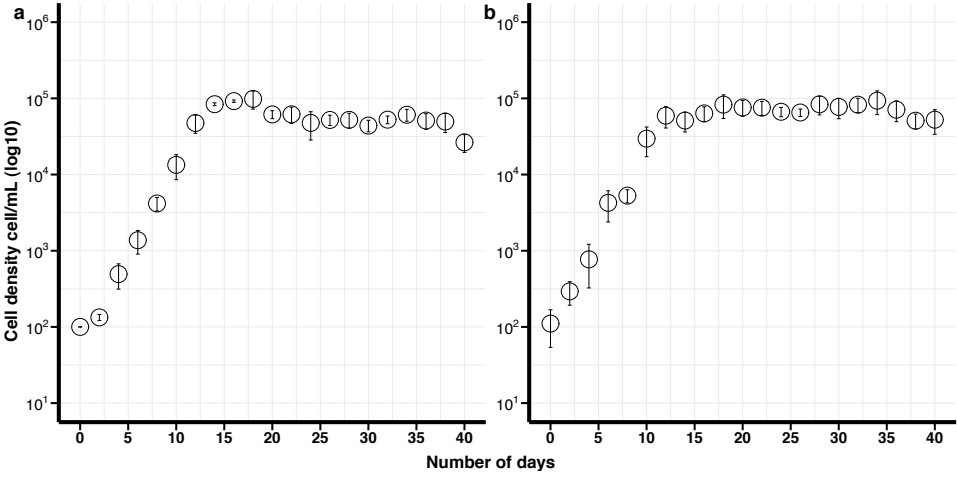

**Figure 2 Growth curves of *Effrenium voratum* culture isolates with standard deviation bars (SD).** (A) Control (non-cryopreserved) culture, (B) cryopreserved culture.

### *Effrenium voratum*

The maximum growth rate for non-cryopreserved *E. voratum*, was 0.87 divisions day$^{-1}$ and the maximum cell density $9.8 \times 10^4$ cells mL$^{-1}$ after 18 days in culture. The maximum growth rate for the cryopreserved cells was 0.76 divisions day$^{-1}$ and the maximum cell density was $9.7 \times 10^4$ cells mL$^{-1}$ after 18 days (Fig. 2). There was no significant difference in the maximum growth rate or the maximum cell density between the cryopreserved and the non-cryopreserved culture isolates (Wilcoxon test; maximum growth rate $p$-value = 0.2101, maximum cell density $p$-value = 0.6761). There were no significant differences in

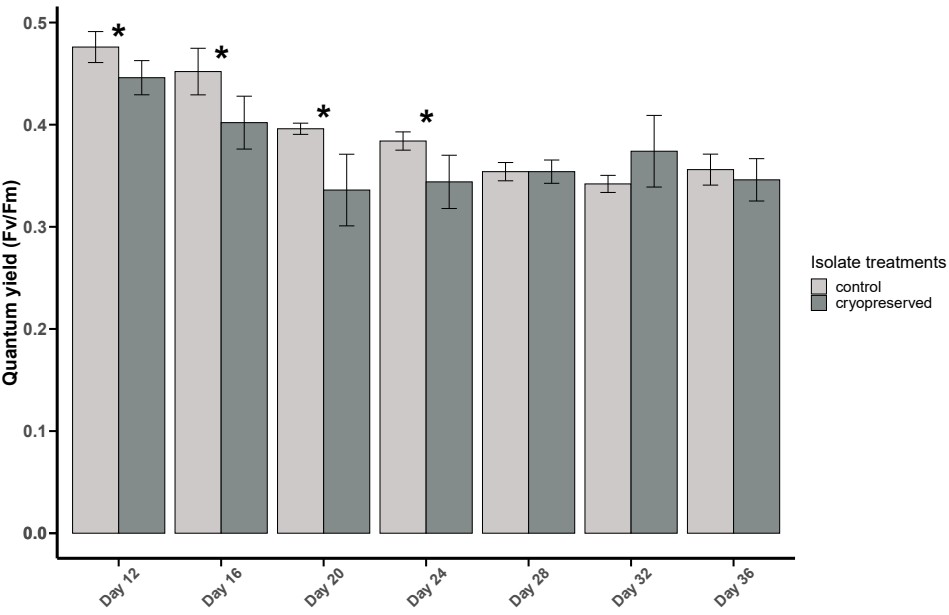

**Figure 3** **Quantum yield ($F_v/F_m$) of *Breviolum psygmophilum* culture isolates at different days of their growth starting from the late exponential to late stationary phase.** Error bars are $\pm$ standard deviation. An asterisk (*) refers to the values that are significantly different ($p$-value $< 0.05$).

the growth rates of the cryopreserved and the non-cryopreserved isolates during the late exponential growth phase ($p$-value $= 0.1437$).

## PAM fluorometry

Due to low cell numbers of the culture isolates at the beginning of a culturing period, the cryopreserved and the control culture isolates were grown to late exponential phase before measurement of photosynthetic parameters to assess the physiology of both the control (non-cryopreserved) and cryopreserved *B. psygmophilum* and *E. voratum* culture isolates.

## Photosystem II (PSII) quantum yield ($F_v/F_m$)

The PSII quantum yield of *B. psygmophilum* culture isolate was significantly higher in the control isolate compared to the cryopreserved isolate on day 12 (control isolate: 0.48 $\pm$ 0.02, cryopreserved treatment: 0.45 $\pm$ 0.02; Wilcoxon test, $p$-value $= 0.0345$; Fig. 3). This significant difference in the quantum yield of the control and the cryopreserved isolates continued from day 12 to day 24, whereas from day 28 to day 36 there were no significant differences in the yield (Fig. 3; Table S1). Linear regression showed a significant negative relationship between culture age and quantum yield in the control and cryopreserved treatments (control isolate $r^2 = 0.8105$, $p$-value $<0.001$ and the cryopreserved isolate $r^2 = 0.2887$, $p$-value $<0.001$; Fig. S2).

There was a significant difference in *E. voratum* PSII quantum yield in the control and the cryopreserved isolates on day 20, but on all the other days there were no significant differences (Fig. 4; Table S1). In both the control and the cryopreserved culture isolates, linear regression showed significant negative relationship between days of the culture and

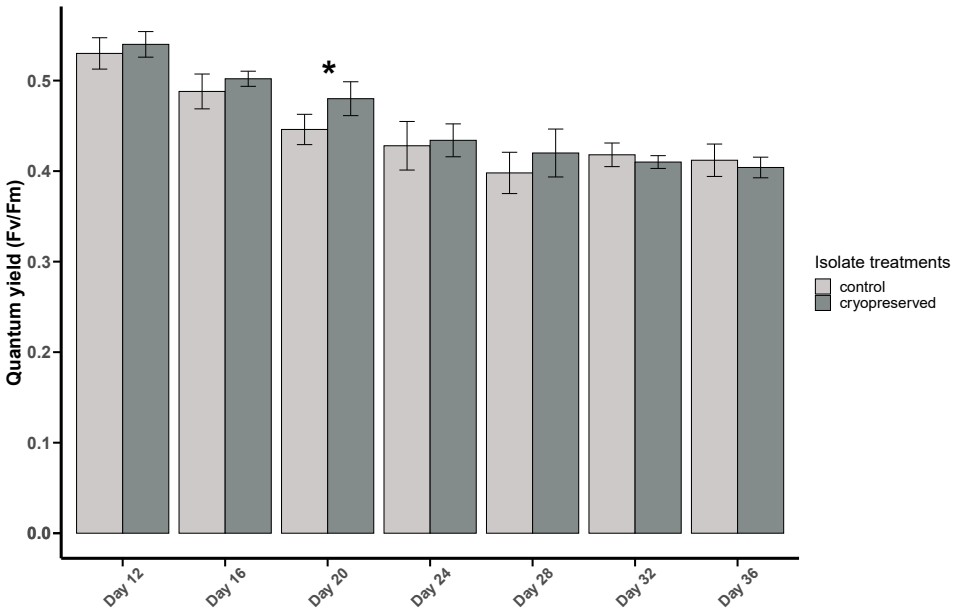

**Figure 4** Quantum yield ($F_v/F_m$) of *Effrenium voratum* culture isolates at different days of their growth starting from the late exponential to late stationary phase. Error bars are $\pm$ standard deviation. An asterisk (*) refers to the values that are significantly different ($p$-value < 0.05).

the quantum yield (control isolate; $r^2 = 0.6259$, $p$-value <0.001 and the cryopreserved isolate; $r^2 = 0.8479$, $p$-value <0.001; Fig. S3).

### Comparison between B. psygmophilum and E. voratum quantum yield

At the start of the PAM analysis, there was a significant difference between the quantum yields of the control and cryopreserved isolates of *B. psygmophilum* and *E. voratum* (*B. psygmophilum* control isolate: 0.48 ± 0.02 and *E. voratum* control isolate: 0.53 ± 0.02; Wilcoxon test, $p$-value = 0.0107 and for the cryopreserved *B. psygmophilum* isolate 0.45 ± 0.02 and *E. voratum* cryopreserved isolate: 0.54 ± 0.01; Wilcoxon test, $p$-value = 0.0107).

### Maximum Electron Transport Rate (ETRmax)

There were no significant differences between the ETRmax of the control and cryopreserved isolates between days 12 to 36 (Fig. 5; Table S2). On day 12 the ETRmax of the control and cryopreserved *B. psygmophilum* isolates were 44.2 ± 2.01 μmol photons m$^{-2}$ s$^{-1}$ and 41.6 ± 1.44 μmol photons m$^{-2}$ s$^{-1}$ respectively (Fig. 5). On the last day of their late exponential phases this had decreased to 19.4 ±2.72 μmol photons m$^{-2}$ s$^{-1}$ and 20.5 ± 4.82 μmol photons m$^{-2}$ s$^{-1}$ (Fig. 5). Linear regression showed a significant negative relationship between the days of the culture and the ETRmax in both culture treatments (control isolate $r^2 = 0.6419$, $p$-value <0.001 and the cryopreserved isolate $r^2 = 0.7126$, $p$-value <0.001; Fig. S4).

There were no significant differences between the ETRmax of the control and cryopreserved isolates between day 12 to day 36 (Fig. 6; Table S2). In *E. voratum*, the ETRmax on day 12 of the control culture isolate was 42.0 ± 1.97 μmol photons m$^{-2}$

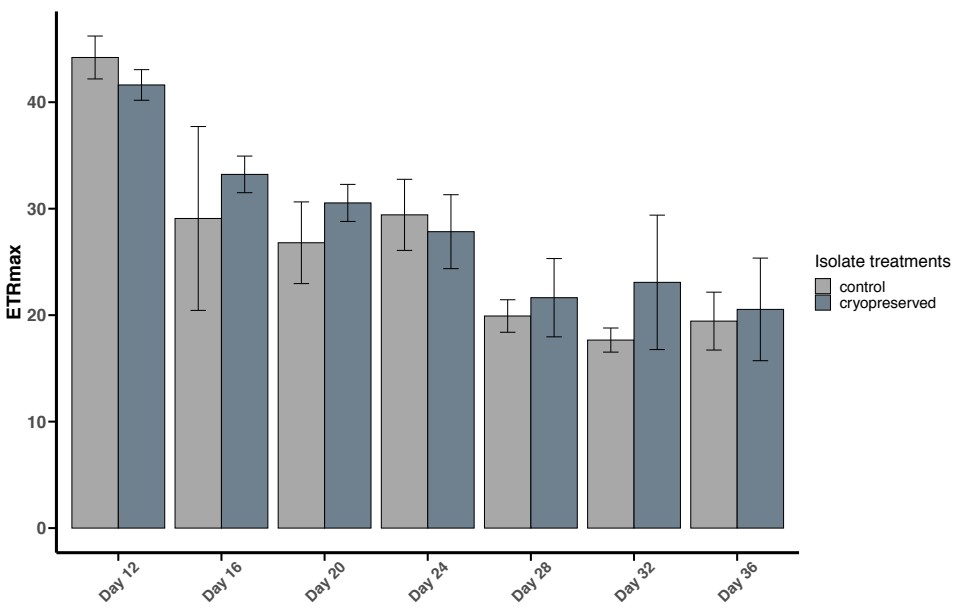

**Figure 5** **Maximum Electron Transport Rate (ETRmax) of *Breviolum psygmophilum* culture isolates at different days from the late exponential to late stationary phase.** Error bars are ± standard deviation.

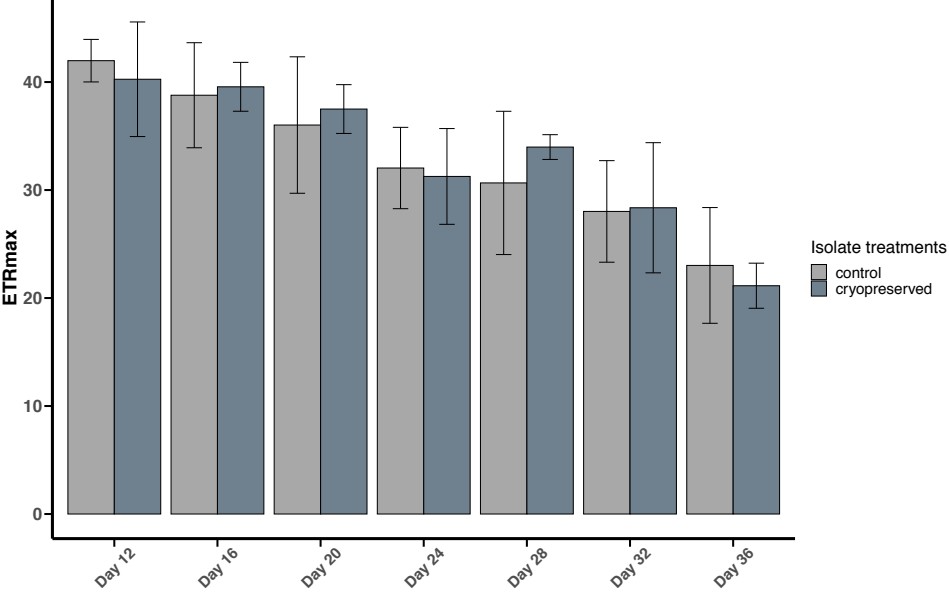

**Figure 6** **Maximum Electron Transport Rate (ETRmax) of *Effrenium voratum* culture isolates at different days of their growth starting from the late exponential to late stationary phase.** Error bars are ± standard deviation.

$s^{-1}$, and the cryopreserved 40.3 ±5.31 μmol photons $m^{-2}$ $s^{-1}$. The ETRmax decreased to 23.0 ± 5.36 μmol photons $m^{-2}$ $s^{-1}$ for the control and 21.1 ±2.09 μmol photons $m^{-2}$ $s^{-1}$ for the cryopreserved isolates (Fig. 6). Linear regression showed a significant negative relationship between day and ETRmax in the control and cryopreserved isolates (control isolate $r^2 = 0.5297$, $p$-value <0.001 and cryopreserved isolate $r^2 = 0.6746$, $p$-value <0.001; Fig. S5).

### Comparison between *B. psygmophilum* and *E. voratum* maximum electron transport rate (ETRmax)

At the start of the experiment, there was no significant difference between the ETRmax of the control and cryopreserved isolates of *B. psygmophilum* and *E. voratum* (*B. psygmophilum* control isolate: 44.2 ± 2.01 and *E. voratum* control isolate: 42.0 ± 1.97; Wilcoxon test, $p$-value = 0.2087 and for the cryopreserved *B. psygmophilum* isolate 41.6 ± 1.44 and *E. voratum* cryopreserved isolate: 40.3 ± 5.31; Wilcoxon test, $p$-value = 1).

## DISCUSSION

In establishing a species bank for the Symbiodiniaceae, cryopreserved microalgal cells should remain viable and be easily grown back into healthy cultures. Long-term preservation of Symbiodiniaceae dinoflagellates will protect them for future scientific research establishing a valuable resource for researchers interested in biological research of *Symbiodinium*–invertebrate symbioses (*Lin et al., 2019*). In this study, an established rapid freezing technique with DMSO as the CPA was applied to the cryopreservation of two Symbiodiniaceae culture isolates from two distinct genera. Both culture isolates had cell survival rates of greater than 50% after freezing, and they were able to grow and divide normally after thawing.

Robust cryopreservation experiments should ensure that microalgal cultures retain their ability to undergo mitosis after thawing (*Gwo et al., 2005*). The cell viabilities of the cryopreserved *B. psygmophilum* and *E. voratum* culture isolates were assessed after a three-day recovery period rather than directly after thawing, to avoid overestimating cell survival (*John, Roland & Erica, 2000*; *Taylor & Fletcher, 1999*; *Tzovenis et al., 2004*). Comparing the maximum growth rates of the control and the cryopreserved treatments of *B. psygmophilum* and *E. voratum*, there were no significant differences. The similar maximum growth rates between control and cryopreserved culture isolates showed that the cells were able to divide and remove the cryoprotectant from their intracellular matrices during the recovery phase (*Kihika et al., 2022a*; *Taylor & Fletcher, 1999*) and that the growth of cryopreserved cultures was not impaired after freezing and thawing.

The low number of cells in culture immediately after thawing prevented the application of PAM fluorometry for assessing cell health due to low fluorescence signal. Both the control and cryopreserved Symbiodiniaceae cells were allowed to grow and divide to the late exponential phase before the start of PAM measurements. From the growth curves data, the late exponential phase occurred around day 12 and both the cryopreserved and control isolates gave the highest value for quantum yield and ETRmax at this point. This is likely because there were many healthy cells actively dividing, and their photosynthetic

performance was highest at that stage of their growth phase. PAM analyses were undertaken every four days to allow investigation of potential differences in photosynthetic activity of the cells between the control and cryopreserved isolates until the late stationary phase (day 36). At this stage cell division was low and most likely due to the shutdown of many biochemical pathways as stationary phase proceeded (*ANACC, 2022*; *Fogg & Thake, 1987*).

The quantum yield of the control *B. psygmophilum* culture isolate was significantly higher on day 12 of the late exponential phase compared to the cryopreserved isolate which might have been due to the cells having stayed active and hence proliferated easily (*Lv et al., 2018*; *Rohit & Venkata Mohan, 2018*) unlike the cryopreserved isolates that had to recover after thawing before dividing again and proliferating under normal growth conditions. The yield of the control isolates remained high from day 12 to day 24 compared to the cryopreserved isolates and this indicates that the control isolate maintained a high photosynthetic activity during that period. This could have been because of the availability of nutrients in the growth media for the cells to photosynthesize and actively divide for those days. During the late stationary phases, there were no significant differences. This could be due to the cultures having similar rates of cell division and death. In *E. voratum*, the quantum yield of the control and the cryopreserved isolates during the late exponential between day 12 and day 16 were not significantly different, apart from day 20 where the yield of the cryopreserved isolate was significantly higher. This shows that the cryopreserved isolate had fully recovered after cryopreservation, and the cells were in good health and maintaining their photosynthetic ability. There were no significant differences in the yield of the culture isolates during the stationary phase indicating that cell metabolic rate and photosynthetic efficiency was not affected by the freezing and thawing experiments. The PSII quantum yield of *E. voratum* in both the control and the cryopreserved treatments was significantly higher than for *B. psygmophilum* culture isolate during the late exponential and stationary phases on day 12 and 36. This observation reflects differences in photosynthetic performances in the two Symbiodiniaceae isolates from separate genera.

Comparing between the overall quantum yield of the free-living (non-symbiotic) *E. voratum* and the symbiotic *B. psygmophilum* during the late exponential growth phase when the PAM analysis was first carried out, there was a significantly higher yield in the *E. voratum* isolate indicating that it had a higher photosynthetic performance than *B. psygmophilum*. The reason is not clear, but it could be that, due to being a non-symbiont, the photosynthetic centres have a high activity that generates enough photosynthates for the cell's metabolism and growth. Being a symbiotic isolate, *B. psygmophilum* might depend on the interaction with its host, but further studies are required to ascertain this (*Klueter et al., 2017*).

The stationary phase (day 36) of the growth curves in both *B. psygmophilum* and *E. voratum* culture isolates had a lower quantum yield than at the end of the exponential phase. This is likely caused by stress due to nutrient limitation in the culture media (*Kromkamp et al., 2008*; *Ramanna et al., 2014*; *White, Anandraj & Trois, 2013*). In many microalgal cultures, decreasing $F_v/F_m$ values represents a decline in photosynthetic efficiency (*White, Anandraj & Bux, 2011*). Alternatively, the Symbiodiniaceae cells may produce excess oxygen through photosynthesis that causes the cells susceptibility to oxidative stress (*Roth,*

_2014_). The production of reactive oxygen species (ROS) such as singlet oxygen ($^1O_2$ *) inside the cells during photosynthesis can cause photo-oxidative damage to the chlorophyll pigments leading to the loss of PSII activity (_Lesser, 2006_; _Ramanna et al., 2014_; _Rehman et al., 2016_).

There were no significant differences in ETRmax between the control and the cryopreserved cultures of both _B. psygmophilum_ and _E. voratum_. These results show that both isolates had similar rates of electron transport through PSII (_Ramanna et al., 2014_; _White, Anandraj & Bux, 2011_), illustrating that cryopreservation did not have any negative effects on the physiology and growth of the cryopreserved cells. The cryopreserved _B. psygmophilum_ had both high cell viability and maximum growth rate compared to _E. voratum_. The reason for this is not clear but it may be that the smaller cell size of _B. psygmophilum_ enables the cells to divide and grow faster due to lower cell energy requirements.

## CONCLUSIONS

The results of this study have shown that cryopreserving the symbiotic _B. psygmophilum_ and non-symbiotic _E. voratum_ Symbiodiniaceae isolates through rapid freezing did not affect the overall photosynthetic health and the growth rates of these dinoflagellates. Overall, the dinoflagellate cells were able to recover fully and divide after thawing and grow back to healthy cultures under normal incubation conditions. The successful cryopreservation of the symbiotic _B. psygmophilum_ isolate provides a foundation for designing future _ex-situ_ conservation strategies including the invertebrate host stony corals and will provide a resource for studying and understanding of the complexities of coral-algal symbiosis. Further research is needed to ensure that cryopreserved cells are still accepted by their symbiotic host, but this was beyond the scope of the current study. Current global climatic changes threaten the existence of both free living and symbiotic Symbiodiniaceae species and adopting cryopreservation as a long-term storage method for these species will protect them for future ecological restoration efforts and research purposes.

## ACKNOWLEDGEMENTS

We thank Jessica Schattschneider for her valuable data analysis advice and Clint Oakley for supplying the Symbiodiniaceae culture isolates from Victoria University of Wellington.

### Funding
This research was supported by funding from the New Zealand Ministry of Business, Innovation and Employment, Contract number: CAWX0902 and a Cawthron Institute Internal Capability Investment Fund scholarship. The funders had no role in study design, data collection and analysis, decision to publish, or preparation of the manuscript.

### Grant Disclosures

The following grant information was disclosed by the authors:
New Zealand Ministry of Business, Innovation and Employment: CAWX0902.
Cawthron Institute Internal Capability Investment Fund.

### Competing Interests

Susanna A. Wood is an Academic Editor for PeerJ.

### Author Contributions

- Joseph K. Kihika conceived and designed the experiments, performed the experiments, analyzed the data, prepared figures and/or tables, authored or reviewed drafts of the article, and approved the final draft.
- Susanna A. Wood conceived and designed the experiments, analyzed the data, prepared figures and/or tables, authored or reviewed drafts of the article, and approved the final draft.
- Lesley Rhodes conceived and designed the experiments, authored or reviewed drafts of the article, and approved the final draft.
- Kirsty F. Smith conceived and designed the experiments, authored or reviewed drafts of the article, and approved the final draft.
- Juliette Butler performed the experiments, authored or reviewed drafts of the article, and approved the final draft.
- Ken G. Ryan conceived and designed the experiments, analyzed the data, prepared figures and/or tables, authored or reviewed drafts of the article, and approved the final draft.

### Data Availability

   The raw data is available in the Supplementary Files.

### Supplemental Information

Supplemental information for this article can be found online at http://dx.doi.org/10.7717/peerj.14885#supplemental-information.

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
