# Peer review of "Assessment of the recovery and photosynthetic efficiency of Breviolum psygmophilum and Effrenium voratum (Symbiodiniaceae) following cryopreservation"

_PeerJ, doi:10.7717/peerj.14885_

## Round 0.1 · original submission · Minor Revisions

Dear Authors

Kindly include all suggestions (comments) given by all reviewers before re-submission of revised version. Thank you.

Editor

·

Basic reporting

No Comment --all necessary basic infromation are reported

Experimental design

Have some questions -- see below in comments

Validity of the findings

no comment

Additional comments

Work by Kihika et al details the effects of cryopreservation on Symbiodiniaceae cells by measuring chlorophyll a fluorescence. The work is interesting, straightforward and simple, However, i don't understand what is the real meaning of this work.

While I don't have any comments on how the manuscript is written - Introduction has all the necessary information to support the story. I do have some, which need to be clarified by the authors

1. The whole basis of this work is to use cryopreservation techniques to potentially preserve Symbiodiniaceae cells for the future, when necessary.

2. I don't understand what is the advantage of using cryopreserved cell over cultured cells?

3. While the authors have shown there is no difference between cell division and growth in control and cryopreserved cells, is the uptake of thawed cells when compared to cultured cells by corals will be same? This is the important question that needs to be addressed, rather than successfully cryopreserving different Symbiodiniaceae species. If corals cannot uptake them and form symbiosis, then no matter how effective is the cryopreservation, it is of no use.

4. Also, for the results, I see that both Fv/Fm and ETR mac are low for both types of cells. So, when using them for infecting corals, will they be viable and successful?

5. The only advantage I see is the work that goes into maintaining the cells in culture and cryopreserved cells.

6. In this work, the cells were cryopreserved for 1 week, what happens if the cells are preserved for say years?

I don't see any advances in this aspect of research as this is not first work to show successful cryopreservation and viability of cells after thawing. The only thing that is new is measurement of photosynthetic efficiency.

Maybe the way the manuscript is written does not give the impression of its importance.

Reviewer 2 ·

Basic reporting

Kihika et al. developed cryopreservation technology for Breviolum psygmophilum and Effrenium voratum. These findings will be interesting for the readers. The manuscript is well written, and they described the purpose clearly. However, following minor points have to address before acceptance.
Abstract:
Line 39: It’s better to use “Many” or “different”. In this sentence both words are indicated same meanings. This reviewer suggest to omit one word.
Introduction:
Line 73-76: This sentence is too big. This reviewer suggest to make two sentence. Also, it’s better to start the sentence as follow-
“Cryopreservation is a technique of preserving viable biological samples………………”
Materials and methods:
Cultures studied and their growing conditions: How many cells were used to culture in the medium?
Rapid freezing technique: This reviewer suggest to mention the rack height during 1st freezing
Results:
Figure 1 & Figure 2: Have to use tick marks in the outside of both “X” and “Y” axis.
Figure 3 - Figure 6: Figure quality is not appropriate. Have to remove square outer line. “X” and “Y” axis should like Figure 1.

Conclusion:
The first sentence is not appropriate for conclusion. This sentence may add in discussion section.

Experimental design

Experimental design is appropriate.

Validity of the findings

Conclusion have to improve.

Reviewer 3 ·

Basic reporting

1. The introduction is in general well written and establishes the background. Relevant literature is cited appropriately.

2. The figures present the data clearly and appear to have appropriate SD whiskers. An apparently complete set of raw data is supplied.

3. The written English is fluent and generally well edited. There are occasional unidiomatic phrasings that do not impede understanding (e.g., line 69: "coral-reefs restoration"; the sentence in lines 193–195 has no verb) and minor errors of punctuation (e.g., "however" should be preceded by a semicolon and followed by a comma; there is an inappropriate semicolon on line 118).

Experimental design

4. The research question is appropriately defined, and the present study fills a defined knowledge gap. The methods could use additional detail to support replicability, especially as regards the cooling and counting protocols.

Validity of the findings

5. The data appear to be robust and statistically sound. Control groups are provided for all experiments. The conclusions drawn from the data are appropriate.

Additional comments

6. Line 92: "PAM fluorometry has not however, been used to characterize photosystem II function in marine algae following cryopreservation." See Hagedorn and Carter 2015, cited in the present manuscript. Is that not an example of this having been done? A claim to novelty in this manuscript may need to be more specific than this.

7. Line 133. Various cryopreservation straws are available and the choice will affect heat transfer. Please specify the straw volume, type, and manufacturer.

8. Line 134. This method will be difficult to replicate closely without additional information. If feasible with available equipment, a measured cooling curve inside a dummy straw filled with solution and cooled with this setup would be helpful. A photograph or diagram of the setup, distance above the liquid nitrogen, etc., would likewise be helpful. Were the straws manually nucleated or left to nucleate spontaneously? Such details can be in supplemental information.

9. Line 141. A minor point to consider: 0.5-mL French straws should thaw completely within 10–15 s under the reported conditions.

10. Line 156. On reading the reference, it is not clear what is meant by "direct-counting method." Please provide some detail on the counting method and equipment used.

11. Line 251: "t quantum yield"?

12. Lines 246, 270, etc.: It is incorrect to have both = and <. Either = or < is appropriate, e.g. "p < 0.001".

13. Line 279: "p-value = p-value"?

14. Line 291: In establishing broader context in this way, it is probably best to avoid self-citation. Self-citation is more appropriate for referring to previous experimental methods and results that are necessary for the present discussion.

15. Line 293: This sentence states the results of the present work. What is the purpose of the citation?

---

## Round 0.2 · accepted · Accept

The authors have included reviewers' suggestions. The manuscript is accepted for publication.

·

Basic reporting

Basic reporting is OK -- clear English, references cited and background

Experimental design

I don't have any more comments

Validity of the findings

I don't have any more comments

Additional comments

I have read this submission and the response of the authors to the comments and suggestions. I do not have any further comments or suggestions

Reviewer 3 ·

Basic reporting

The authors have responded thoughtfully and effectively to the comments. This reviewer has no additional comments and considers the manuscript suitable for publication.

The reviewer thanks the authors for annotating the edited manuscript with comments linking changes to specific reviewer points. This was a useful addition.

Experimental design

No comment

Validity of the findings

No comment